# AdaptiveISP: Learning an Adaptive Image Signal Processor for Object Detection

**Yujin Wang** [1]   **Tianyi Xu** [1,2]   **Fan Zhang** [1]   **Tianfan Xue** [3,1]   **Jinwei Gu** [3]

[1] Shanghai AI Laboratory          [2] Peking University
{wangyujin, zhangfan}@pjlab.org.cn, photon@stu.pku.edu.cn
[3] The Chinese University of Hong Kong
{tfxue@ie, jwgu@cse}.cuhk.edu.hk

## Abstract

Image Signal Processors (ISPs) convert raw sensor signals into digital images, which significantly influence the image quality and the performance of downstream computer vision tasks. Designing ISP pipeline and tuning ISP parameters are two key steps for building an imaging and vision system. To find optimal ISP configurations, recent works use deep neural networks as a proxy to search for ISP parameters or ISP pipelines. However, these methods are primarily designed to maximize the image quality, which are sub-optimal in the performance of high-level computer vision tasks such as detection, recognition, and tracking. Moreover, after training, the learned ISP pipelines are mostly fixed at the inference time, whose performance degrades in dynamic scenes. To jointly optimize ISP structures and parameters, we propose AdaptiveISP, a task-driven and scene-adaptive ISP. One key observation is that for the majority of input images, only a few processing modules are needed to improve the performance of downstream recognition tasks, and only a few inputs require more processing. Based on this, AdaptiveISP utilizes deep reinforcement learning to automatically generate an optimal ISP pipeline and the associated ISP parameters to maximize the detection performance. Experimental results show that AdaptiveISP not only surpasses the prior state-of-the-art methods for object detection but also dynamically manages the trade-off between detection performance and computational cost, especially suitable for scenes with large dynamic range variations. Project website: https://openimaginglab.github.io/AdaptiveISP/.

## 1   Introduction

Image Signal Processors (ISPs) play a fundamental role in camera systems. Originally, ISPs aimed at enhancing the perceptual quality, focusing on photography-related applications. Recently, machine vision cameras also optimized the ISP pipeline for downstream recognition tasks [35, 39, 34, 36]. For recognition tasks, studies have shown a specially designed ISP can significantly enhance their performance. However, most machine vision cameras [13] still prefer a static hand-designed ISP and manually-tuned parameters, making it sub-optimal for downstream recognition tasks and also inflexible for dynamic scenes. Another solution is to directly train detection networks that take raw files as input, skipping the entire ISP processing. However, this may require re-training the detection network for each camera sensor, as the raw format varies between cameras, and studies [23] also show that the raw detection network still performs worse than the detection network runs on ISP-processed images. Therefore, a well-tuned ISP is important for downstream recognition tasks.

In this work, we aim to design an ISP that can dynamically adapt its pipeline and parameters for different inputs, tailored for a given high-level computer vision task. This problem faces two challenges: complexity and efficiency. First, it is a complicated optimization problem to jointly

38th Conference on Neural Information Processing Systems (NeurIPS 2024).

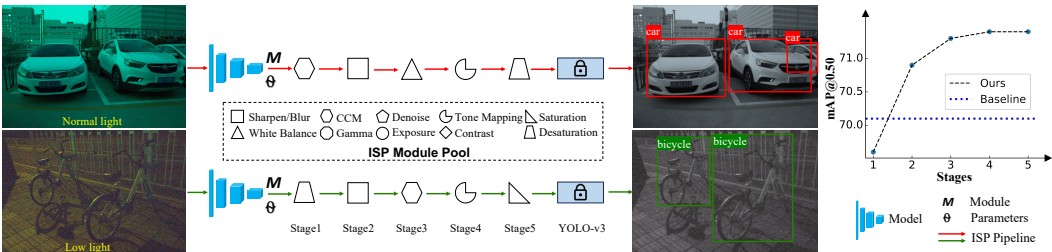

Figure 1: AdaptiveISP takes a raw image as input and automatically generates an optimal ISP pipeline $\{M_i\}$ and the associated ISP parameters $\{\Theta_i\}$ to maximize the detection performance for any given pre-trained object detection network with deep reinforcement learning. AdapativeISP achieved mAP@0.5 of 71.4 on the dataset LOD dataset, while a baseline method [35] with a fixed ISP pipeline and optimized parameters can only achieve mAP@0.5 of 70.1. Note that AdaptiveISP predicts the ISP for the image captured under normal light requires a CCM module, while the ISP for the image captured under low light requires a Desaturation module.

re-organize the ISP modules, update their parameters, and also improve the performance of the downstream recognition module. Because of this complexity, previous works only update the parameters [35, 25, 30, 31, 38, 16, 26]. Some recent works jointly optimize the ISP pipeline and parameters [10, 28], but they are not designed for downstream recognition tasks. Second, ISP optimization must be efficient enough to adapt to dynamically changing scenes, which is particularly important in real-time applications such as autonomous driving and robotics. The majority of ISP pipeline optimization methods use searching strategies, such as the neural architecture search (NAS) method [39] and the multi-object optimization search method [34], which often takes several hours, making them infeasible for real-time applications across dynamically changing scenes.

To solve this challenge, we observe that the majority of input images only require a few ISP operations to increase the downstream recognition accuracy. As shown in Figure 1 on the right, only the first two stages of ISP already boost the detection accuracy (mAP) from 67.8 to 70.9, and the rest of the three stages only further boost it to 71.4. Only challenging examples require more complicated pipelines. Therefore, we can model the ISP configuration process as a Markov Decision Process [29], and our AdaptiveISP only selects one ISP module at each stage, as shown in Figure 1 on the left. This greatly reduces the search space at inference time and enables adaptively changing the ISP length, spending less time on easy inputs.

Based on this idea, we introduce AdaptiveISP, a real-time reconfigurable ISP based on reinforcement learning (RL). As shown in Figure 1, AdaptiveISP takes a linear image as input and generates an optimal ISP pipeline with associated parameters that best fit this image for object detection. Unlike previous neural architecture search (NAS) [39] that generates the entire pipeline at once, AdaptiveISP takes a greedy approach to only generate one module at each iteration, greatly reducing the searching time. At each iteration, a lightweight RL agent takes the processing output from the previous stage as input and finds the optimal module for the next iteration. Because of its efficiency, AdaptiveISP only takes 1 ms to predict one stage and can generate different pipelines for different scenes on the fly in real-time, as illustrated in Figure 1.

Furthermore, we design a reinforcement learning scheme tailored for ISP configuration. First, we integrate a pre-trained fixed object detection network, YOLO-v3 [32], into the optimization system as a loss function, guiding our model to prioritize specific high-level computer vision tasks. Second, with the observation that many later-stage ISP modules may only bring little improvement to detection, we introduce a new cost penalty mechanism so that AdaptiveISP supports dynamically trading off object detection accuracy and ISP latency.

The proposed AdaptiveISP have several distinctive properties compared to a typical ISP for visual quality. First, ISP for detection can be much simpler. As shown on the right side of Figure 1, with only 4 stages, it achieves the best detection accuracy, while traditional ISP may need more than 10 stages to enhance image quality. Second, while color processing is important in both types of ISPs for both detection and viewing, their behaviors are different. For instance, ISPs for detection often completely desaturate color in low-light scenarios to boost the detection rate, as shown in the second-row of Figure 1, which rarely happens for traditional ISPs. Third, a simple sharpening or

blurring module may greatly enhance detection accuracy, while a more widely used denoising module in traditional ISPs is not super helpful for detection.

We have evaluated the proposed AdaptiveISP on the LOD [9], OnePlus [39], and synthetic raw COCO [19] datasets and showed that it outperforms prior state-of-the-art methods under different challenging conditions and different downstream tasks. Experimental results also demonstrate the ability to dynamically switch from a high-accuracy ISP pipeline to a low-latency one.

## 2   Related Work

**ISP Parameter Tuning.** Recent studies have explored various methods for optimizing Image Signal Processing (ISP) hyper-parameters, particularly those based on handcrafted designs and tailored to meet the demands of downstream evaluation metrics. One category of methods focuses on derivative-free optimization techniques. Nishimura *et al.* [26] introduced an automatic image quality tuning method that employs nonlinear optimization and automatic reference generation algorithms. Another category utilizes gradient-based optimization techniques. Tseng *et al.* [35] introduced a gradient optimization method that relies on differentiable approximations, allowing for efficient hyper-parameter tuning but the ISP parameters are fixed during the inference stage. Immediately afterward, some researchers realized that one set of parameters was not necessarily suitable for different scenarios. Qin *et al.* [30] proposed an attention-based CNN method, but it does not consider sequence-specific prior knowledge. Then, Qin *et al.* [31] proposed a sequential ISP hyper-parameter prediction framework, which optimizes the ISP parameters by leveraging sequential relationships and parameter similarities. Additionally, Liu *et al.* [22] proposed the IA-YOLO approach, which can adaptively process images under both normal and adverse weather conditions. Yoshimura *et al.* [38] proposed *DynamicISP*, which can causally and smoothly control the parameters of the current frame according to the recognition result of the previous frame. Departing from approximation methods, Mosleh *et al.* [25] introduced a hardware-in-the-loop approach. This approach directly optimizes hardware-based image processing pipelines to meet specific end-to-end objectives by using a novel 0th-order stochastic solver. Furthermore, an image enhancement method based on reinforcement learning, proposed by Kosugi *et al.* [16], leverages reinforcement learning techniques to optimize hyper-parameters in an unpaired manner.

**ISP Pipeline Design.** The conventional ISP pipelines are crafted to adhere to human visual perception, and this alignment might not always be conducive to fulfilling the demands of downstream high-level tasks. The prior research works [34, 39] have proved that handcrafted ISP configuration does not necessarily benefit the downstream high-level vision tasks. ReconfigISP [39] proposed a novel Reconfigurable ISP whose architecture and parameters can be automatically tailored to specific data and tasks by using neural architecture search (Darts) [20]. RefactoringISP [34] jointly optimized ISP structure and parameters with task-specific loss and ISP computation budgets through multi-objective optimization algorithm (NSGA-II) [4]. However, these approaches maintain the ISP pipeline and parameters fixed during inference, regardless of the distinct characteristics of the input. An innovative study addresses the challenge of photo retouching by leveraging deep learning on unpaired data, allowing users to emulate their preferred retouching style [10]. Additionally, recent studies have explored replacing traditional ISP pipelines or modules with deep learning models to enhance image quality. PyNET [11] proposed a unified model that directly learns the RAW-to-RGB mapping to improve mobile photography, while CycleISP [40] introduced a noise-aware denoising approach. ParamISP [14] developed forward and inverse ISP models conditioned on camera parameters, enabling a more accurate emulation of real-world ISP behavior to enhance image processing performance. DualDn [17] further employs a differentiable ISP to improve denoising capabilities. Although these methods improve image quality, they do not consistently translate into better performance for downstream high-level vision tasks.

## 3   Our Method

Image Signal Processors (ISPs) often consist of a pipeline of image processing modules that primarily transform raw sensor pixel data into RGB images suitable for human viewing [2]. A typical camera ISP pipeline includes two parts: raw-domain and RGB-domain. Compared to raw-domain processing, the RGB-domain processing is normally image-dependent and requires more dedicated design and tuning efforts. Details of ISP pipeline and modules can be found in the *appendix*.

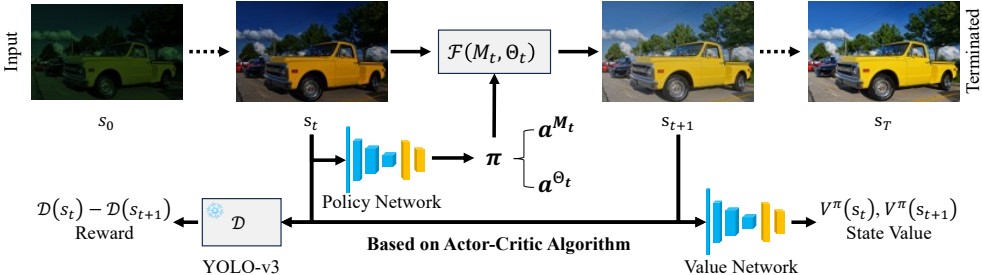

Figure 2: Overview of our method. The ISP configuration process is conceptualized as a Markov Decision Process, where a CNN-based policy network predicts the selection of ISP modules and their parameters. Concurrently, a CNN-based value network estimates the state value. The YOLO-v3 [32] is employed to calculate the reward for the current policy. The entire system is optimized using the actor-critic algorithm [15, 24].

In this work, we focus on the RGB-domain processing. We assume that the captured raw sensor data is already converted to linear RGB images using a simple static raw-domain processing, and our tuning mainly focuses on sRGB-domain processing, similar to [9, 38].

### 3.1 Problem Formulation of AdaptiveISP

Given an input linear RGB image $I$, the goal of AdaptiveISP is to find an optimal ISP pipeline and parameters, with which the processed output image will result in high performance for object detection. Let $\mathcal{F}_{(M,\Theta)}$ denote one ISP module, where $M$ is the ISP module type and $\Theta$ is the set of parameters for that module. An ISP pipeline consists of a sequence of ISP modules $\left\{\mathcal{F}_{(M_t,\Theta_t)}\right\}_{t=0}^{T}$, which transforms an input image $I$ to an output image $I_t$ with $t$ stages of ISP processing as

$$I_t = \left(\mathcal{F}_{(M_t,\Theta_t)} \circ \cdots \circ \mathcal{F}_{(M_0,\Theta_0)}\right)(I). \tag{1}$$

The goal of AdaptiveISP is to predict an optimal ISP pipeline (with $T$ stages) and its parameters, i.e., $\{M_t, \Theta_t\}_{t=0}^{T}$ for an input image, in order to maximize its performance for a given computer vision task (e.g. object detection), which can be formulated as

$$\{M_t, \Theta_t\}_{t=0}^{T*} = \underset{\{M_t,\Theta_t\}_{t=0}^{T}}{\text{argmin}} \ \{\mathcal{D}(I_T)\}, \tag{2}$$

where $\mathcal{D}$ is the detection error with a given object detector. This optimization problem can thus be modeled as a Markov Decision Process [29], which can be solved efficiently via deep reinforcement learning.

### 3.2 Optimization Objectives

Our formulation is similar to [5, 6, 10, 28]. Let us denote the problem as $P = (S, A)$, where $S$ is a state space and $A$ is an action space. Specifically, in our task, $S$ is the space of images, which includes the input images and all the intermediate results in the ISP process, while $A$ is the set of all ISP modules. Since each action includes the selection of ISP modules and the prediction of ISP module parameters, we can decompose the action space $A$ into two parts: a discrete selection of ISP modules $a^M$ and a continuous prediction of ISP module parameters $a^\Theta$. At stage $i$, with a selected ISP module and its parameters $(a_i^M, a_i^\Theta)$, the input image at state $s_i$ is mapped to state $s_{i+1}$. Applying a sequence of $T$ selected ISP modules to an input image corresponds to a trajectory $\tau$ of states and actions:

$$\tau = \left(s_0, a_0^M, a_0^\Theta, \ldots, s_{T-1}, a_{T-1}^M, a_{T-1}^\Theta, s_T\right), \tag{3}$$

where $s_T$ is the stopping state. Our goal is to find a policy $\pi$ that maximizes the accumulated reward during the decision-making process. The policy $\pi$ consists of two sub-policies $(\pi^M$ and $\pi^\Theta)$, where $\pi^M$ takes a state and returns a probability distribution over ISP modules, and $\pi^\Theta$ predicts the parameters $a^\Theta$ of the selected ISP module. In this paper, the reward function with the $i$-th action (i.e., corresponding to the $i$-th stage of ISP processing) is thus written as:

$$r(s_i, a_i^M, a_i^\Theta) = \mathcal{D}(s_i) - \mathcal{D}(s_{i+1}), \tag{4}$$

where $s_{i+1} = p(s_i, a_i^M, a_i^\Theta)$, and $\mathcal{D}$ is the error of object detection. Given a trajectory $\tau$, we define the return $g_t$ as the summation of the discounted rewards after $s_t$:

$$g_t = \sum_{k=0}^{T-t} \gamma^k \cdot r(s_{t+k}, a_{t+k}^M, a_{t+k}^\Theta), \tag{5}$$

where $\gamma \in [0, 1]$ is a discount factor that places greater importance on rewards in the near future. We can thus define the value of state function $V^\pi(s)$ as:

$$V^\pi(s) = \mathop{\mathbb{E}}_{\tau \sim \pi} \left[g_0 | s_0 = s\right], \tag{6}$$

and the value of action function $Q^\pi(s, a^M, a^\Theta)$ as:

$$Q^\pi(s, a^M, a^\Theta) = \mathop{\mathbb{E}}_{\tau \sim \pi} \left[g_0 | s_0 = s, a^M, a^\Theta\right]. \tag{7}$$

Our goal is to select a policy $\pi = (\pi^M, \pi^\Theta)$ that maximizes the expected accumulated reward during the decision-making process:

$$J(\pi) = \mathop{\mathbb{E}}_{s \sim S_0} \left[V^\pi(s)\right], \tag{8}$$

where $S_0$ denotes the entire image dataset.

Similar to [10, 28, 16, 5], we employ deep neural networks to approximate the value function $V^\pi(s)$ and the policy $\pi$. As shown in Figure 2, convolutional neural networks (CNNs) and a fully-connected layer are used as the policy network, which maps the image $s$ into action probabilities $\pi^M(s, \Phi_M)$ (after softmax) and ISP module parameters $\pi^\Theta(s, \Phi_\Theta)$ (after $\tanh$), where $(\Phi_M, \Phi_\Theta)$ are the network parameters. The value network $V^\pi(s, \Phi_V)$ uses a similar architecture with parameters $\Phi_V$. By maximizing the objective $J(\pi)_\Phi$ with training these two networks $\Phi = (\Phi_M, \Phi_\Theta, \Phi_V)$, we can learn to predict the optimal policy $\pi(s)$ for an input image $s$. Specifically, to train the policy network and the value network, as shown in Figure 2, we apply the actor-critic algorithm [15, 24], where the actor is represented by the policy network and the critic is the value network. Details of network architectures and training are provided in the *appendix*.

## 3.3 Implementation Details

To ensure stable and effective reinforcement learning for the ISP pipeline and parameter prediction, we augment the network input and implement several penalty functions to better constrain the training process. Specifically, we modify the reward function Equation 4 to:

$$r(s_i, a_i^M, a_i^\Theta) = \mathcal{D}(s_i) - \mathcal{D}(s_{i+1}) - P_i, \tag{9}$$

where $P_i$ stands for the penalties as explained below.

**Exploitation and Exploration.** To avoid the same ISP module being selected consecutively multiple times by the policy network, we augment the network input with the module usage record (which is represented as $N$ channels where $N$ is the size of the ISP module pool) and one additional "stage" channel. At each stage, if an ISP module is being used, the corresponding "use" channel and penalty of reusing will be set to 1 and 0 otherwise. The "stage" channel is set with the index of the stage. During training, these additional inputs and the penalty of reusing constrain the policy network picking the same ISP module no more than once, which can effectively narrow down the solution space and improve the performance. Details of network input are provided in the *appendix*.

In addition, we want to encourage the policy network $\pi^M(s)$ to explore different ISP modules to prevent the parameter prediction network from insufficient learning. Specifically, we introduce a penalty term $P_e$ on the output entropy of the policy network $\pi^M(s)$ to ensure that the action distribution is not overly concentrated.

$$P_e = \lambda_e \sum_{m \in M} p(m) \log p(m), \tag{10}$$

where $p(m)$ is the probability of an ISP module $m$ in the softmax output, and $\lambda_e \in [0, 1]$ is a penalty coefficient and gradually decreases from 1 to 0 with the progress of the training process.

**Penalty of Computational Time.** ISP pipelines typically consist of multiple modules responsible for specific image processing tasks, such as denoising, sharpening, white balance, and more. These modules exhibit varying computational costs, some being faster and others slower, as shown in Table 4. Consequently, when designing ISP pipelines, it is crucial to consider the distinct computational costs associated with these modules. Particularly, in scenarios requiring swift responses like autonomous driving, optimizing the computational efficiency of ISP pipelines becomes paramount. In order to make our method automatically select the appropriate modules and their sequence, we should account for the computational time of each module during the design of the ISP pipeline. Specifically, we collect the run time of all modules, marked as $\mathbb{M}_c$, which can be found in Table 4, and the penalty of cost $P_c$ is defined as:

$$P_c = \lambda_c \sum_{m \in M} \mathbb{I}_m \cdot \mathbb{M}_c, \tag{11}$$

where $\mathbb{I}_m$ represents the one-hot encoding whether the module $m$ is used, and $\lambda_c$ is the penalty coefficient.

## 4 Experiments

**Datasets.** In line with prior research [9, 27, 38, 39, 30, 31], we train and evaluate our models on widely used real low-light detection datasets and synthetic normal-light datasets, including:

- **LOD.** LOD Dataset [9] is a real-world low-light object detection dataset, which contains 2,230 14-bit low-light raw images with eight categories of objects. This dataset aims to systematically assess the low-light detection performance. There are 1,830 data pairs for training and 400 data pairs for validation. Additionally, it provides accompanying metadata, including ISO, shutter speed, aperture settings, and more, which greatly facilitates our experimental analysis.

- **OnePlus.** OnePlus Dataset [39] is a real-world low-light object detection dataset collected by the OnePlus 6T A6010 smartphone at driving scenes, which contains 141 raw images with three classes of objects in the street scenes. There are 50 pairs for training and 91 pairs for validation. Given the limited size of the dataset, all raw images were utilized as the validation set in our experiments.

- **Raw COCO.** COCO [19] is a large-scale object detection, segmentation, and captioning dataset. To evaluate the generalization ability of our method, we convert the COCO validate dataset (5,000 images) to a synthetic raw-like dataset as our evaluate dataset by using UPI [1], similar with [9, 38, 30, 31].

**Experiment Details.** Similar to [39], YOLOv3 [32] is utilized as the detection model in all methods unless explicitly stated otherwise. YOLOv3 is a robust and fast object detection algorithm. Its real-time applicability, speed, and high-performance capabilities make it a powerful choice for various applications in numerous classic object detection algorithms [4, 33, 8, 21, 18, 5]. In terms of back-propagation, YOLOv3 passes gradients back to our method for optimizing both structure and parameters. It is important to note that the pre-trained YOLOv3 model remains unaltered throughout the training process. During training and inference, we follow [9] to use a fixed input resolution of $512 \times 512$. In addition, unless otherwise stated, our methods and comparison methods are all experiments conducted on handcrafted ISP.

**Evaluation Metrics.** We utilize the mean Average Precision (mAP) across all Intersection over Union (IoU) thresholds to assess the performance of our method, following a similar approach as used in object detection algorithms [4, 33, 8, 21, 18, 5].

### 4.1 Results

**Results on LOD Dataset.** The LOD dataset provides JPEG images with accompanying metadata, which serves as a convenience for the analysis of our experimental results. On this dataset, we choose several baseline methods for evaluation. First, we select two network-based ISP methods, namely Crafting [9] and NeuralAE [27]. Also, we evaluate our method against the static handcrafted pipeline and parameter methods, specifically, Hyperparameter Optimization [35]. Furthermore, we compare our approach with static handcrafted pipelines combined with dynamic parameters techniques, such

Table 1: Experimental results of LOD [9], OnePlus [39], and raw COCO Dataset [19].

| Methods | LOD | | | All OnePlus | | | Raw COCO | | |
|---|---|---|---|---|---|---|---|---|---|
| | mAP @0.5 | mAP @0.75 | mAP @0.5:0.95 | mAP @0.5 | mAP @0.75 | mAP @0.5:0.95 | mAP @0.5 | mAP @0.75 | mAP @0.5:0.95 |
| Crafting [9] | 67.9 | 49.0 | 44.7 | - | - | - | - | - | - |
| NeuralAE [27] | - | - | 45.5 | - | - | - | - | - | - |
| DynamicISP [38] | - | - | 46.2 | - | - | - | - | - | - |
| Hyperparameter Optimization [35] | 70.1 | 49.7 | 46.1 | 69.8 | 48.7 | 43.8 | 53.8 | 38.7 | 36.6 |
| Attention-aware Learning [30] | 70.9 | 51.0 | 46.6 | **70.9** | 48.9 | 44.7 | 53.6 | 38.7 | 36.6 |
| ReconfigISP [39] | 69.4 | 49.6 | 45.6 | 65.1 | 42.1 | 40.4 | 52.6 | 38.0 | 35.8 |
| Refactoring ISP [34] | 68.3 | 47.6 | 44.1 | 66.7 | 44.3 | 40.9 | 52.4 | 38.0 | 35.7 |
| **Ours** | **71.4** | **51.7** | **47.1** | 70.1 | **49.7** | **45.0** | **54.9** | **40.1** | **37.7** |

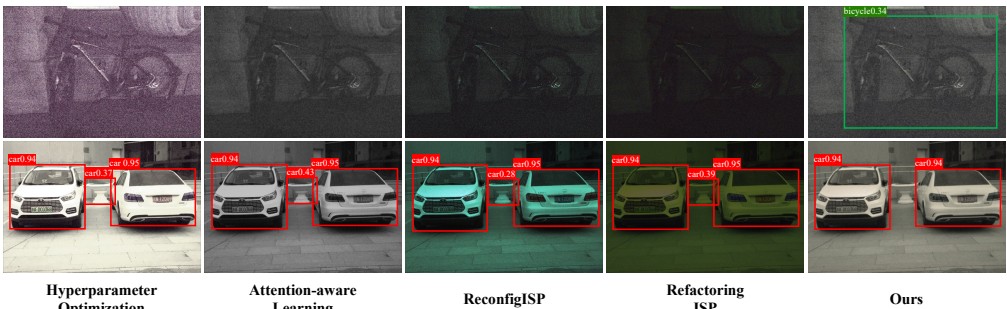

Figure 3: Object detection visualization results on LOD dataset. Our method outperforms the state-of-the-art methods [35, 30, 39, 34] in terms of missed detection and false detection. The methods with fixed pipelines or fixed parameters struggle to effectively handle varying noise levels and brightness scenarios.

as Attention-aware Learning [30] and DynamicISP [38]. Finally, we compare our method with the optimization method for ISP pipelines and parameters, as exemplified by ReconfigISP [39] and Refactoring ISP [34]. ReconfigISP utilizes a neural architecture search algorithm (Darts) [20], while Refactoring ISP is based on Non-dominated Sorting Genetic Algorithm II (NSGA-II) [4].

As shown in the first column of Table 1, our method achieves the best performance across all object detection metrics on the LOD dataset. Note that static pipelines with dynamic parameter methods outperform static pipeline and parameter approaches, dynamic pipelines and parameters yield superior results across all methods. We further show the detection results of our method in Figure 3, showcasing its superior performance in terms of both missed detection and false detection compared to all other methods.

**Cross Datasets Test.** To evaluate the generalization ability of our method on different datasets, we utilize a model trained on the LOD dataset and conduct testing on both the OnePlus and raw COCO datasets. As displayed in the central and rightmost column of Table 1, our method achieves the best performance, which verifies that our method also has the best generalization ability. In terms of the mAP@0.75 evaluation metric, our method exhibits an improvement of approximately 1 point compared to alternative approaches on both Oneplus and raw COCO datasets. Furthermore, our method outperforms Refactoring ISP [34] and ReconfigISP [39] by 2 points on the raw COCO dataset and by 4 points on the OnePlus dataset in terms of mAP@0.5:0.95 evaluation metrics.

**Cross Detectors Test.** To evaluate the generalization ability of our method on different detectors, we use the detection results from the RGB (existing ISP) as a baseline and conduct comparative experiments on DDQ [41], representing the Transformer-based approach, and YOLOX [7], representing the CNN-based approach. As shown in Table 3, all detectors using our AdaptiveISP demonstrate improved detection performance, demonstrating that our method does not overfit one detector, but is suitable for other detectors. Note that DDQ [41] and YOLOX [7] are not used in the training process, but our ISP can still generalize to these detectors at testing time.

**Results for Image Segmentation.** To show our approach can generalize to other tasks, in Table 2, we also conduct an experiment using image segmentation as the downstream task. Note that all models are trained on the LOD dataset using the pre-trained YOLOv3 [32] detector and evaluated

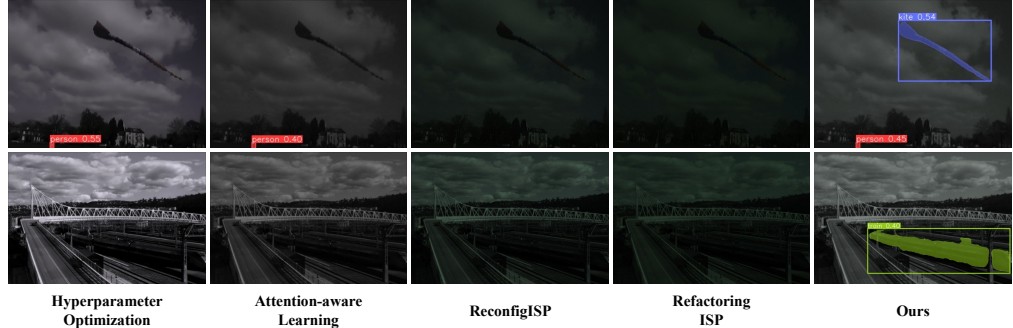

| Hyperparameter Optimization | Attention-aware Learning | ReconfigISP | Refactoring ISP | Ours |

Figure 4: Image segmentation visualization results on raw COCO dataset. Our method detects all the object, while the state-of-the-art methods [35, 30, 39, 34] may miss some.

Table 2: Image Segmentation results on raw COCO datasets [19].

| Methods | mAP@0.5 | mAP@0.5:0.95 |
| --- | --- | --- |
| Hyperparameter Optimization [35] | 46.4 | 28.4 |
| Attention-aware Learning [30] | 45.5 | 27.9 |
| ReconfigISP [39] | 42.1 | 25.2 |
| Refactoring ISP [34] | 40.6 | 24.7 |
| Ours | **47.0** | **28.8** |

on the synthetic raw COCO datasets with the pre-trained YOLO-v5 [12] segmentor. Our method performs better than all baselines, further showing its generalizability for different downstream tasks and algorithms. We further present the segmentation results of our method in Figure 4, highlighting its superior performance in terms of missed detection compared to all other methods.

**Accuracy-Efficiency Trade-off.** Our approach can control $\lambda_c$ to regulate whether the optimization process also takes the computational time of each ISP module into consideration, where $\lambda_c$ is the weight computational cost. By tuning up $\lambda_c$, we can generate a more efficient ISP with a minor drop of the recognition accuracy.

We conducted analysis experiments on the LOD dataset, where $\lambda_c = 0.0$ represents the accuracy-oriented results and $\lambda_c = 0.01$ stands for efficiency-oriented results. We calculated the proportions of the occurrence of each module in the test results. As shown in Table 4, the efficiency-oriented method has a significant reduction in the average running time for each sample, which is accompanied by a slight decrease in performance. In addition, the frequency of appearance of modules with higher computational time, such as Sharpen/Blur and Tone Mapping, decreased by more than 50%. Conversely, modules with lower computational time, like Exposure and White Balance, saw a substantial increase in selection frequency.

**Runtime.** To verify the practicality of our approach, we conducted speed tests using the NVIDIA GTX1660Ti GPU, which offers a computational capability of 11 TOPS—markedly lower than that of the NVIDIA DRIVE Orin™ SoC at 254 TOPS. Our method only takes 1.2 ms per stage during inference, this efficiency is attributed to just need to utilize the light-weight policy network to predict the modules and parameters during inference.

### 4.2   Ablation Study and Analysis

**Adaptive ISP.** We perform a study to show that different data require different ISP pipelines to achieve the best performance. To accomplish this, we begin by gathering the various ISP pipelines predicted by our method on the LOD dataset. Next, we select the three most representative ISP pipelines along with their corresponding input raw images, resulting in three subsets of the LOD dataset. The specifics of these three distinct ISP pipelines can be found in Figure 5. Finally, we conduct cross-testing on the three sets of ISP pipelines generated by our method using the three subsets, as depicted in Figure 5. By analyzing the experimental results, we can see that only the

Table 3: Experimental results of different detectors on LOD dataset. Note that the DDQ [41] and YOLOX [7] do not participate in our training process. All detectors using our method demonstrate improved detection performance.

| Detectors | Methods | mAP@0.5 | mAP@0.75 | mAP@0.5:0.95 |
|---|---|---|---|---|
| YOLO-v3 [32] | RGB | 55.6 | 40.9 | 37.4 |
| | **Ours** | **71.4** | **51.7** | **47.1** |
| YOLOX [7] | RGB | 57.0 | 43.5 | 39.2 |
| | **Ours** | **69.7** | **51.4** | **47.2** |
| DDQ [41] | RGB | 50.3 | 42.0 | 35.9 |
| | **Ours** | **74.0** | **58.1** | **52.0** |

| LOD | Subset 1 | | Subset 2 | | Subset 3 | |
|---|---|---|---|---|---|---|
| Representative Images |  | |  | |  | |
| Metrics | mAP@0.5 | mAP@0.75 | mAP@0.5 | mAP@0.75 | mAP@0.5 | mAP@0.75 |
| Pipeline 1 △→□→○→↺→◺ | **63.0** | **39.8** | 82.1 | 66.8 | 78.2 | 64.5 |
| Pipeline 2 ○→□→↺→△→◺ | 62.1 | 39.6 | **82.7** | **68.9** | **79.2** | 65.2 |
| Pipeline 3 ○→□→△→↺→◺ | 62.4 | 39.5 | 82.4 | 67.8 | 79.1 | **65.4** |

□ Sharpen/Blur  △ White Balance  ○ CCM  ↺ Tone Mapping  ◺ Saturation  ◻ Desaturation

Figure 5: The cross-validated result of different ISP pipelines and its sub-dataset on LOD datasets. Only the most matching pipeline can achieve the best results, which proves that a different pipeline is necessary.

most matching pipeline can achieve the best results, otherwise, it will lead to varying degrees of performance degradation, which also proves that there are different pipeline requirements in different scenarios.

**Module preferences of different images.** Furthermore, we analyze the reasons why different ISP pipelines are necessary for various scenarios. The LOD dataset provides JPEG images with accompanying metadata, which serves as a convenience for the analysis of our experimental results.

It is noticed that there are two different choices at the first stage of the three ISP pipelines: CCM and Desaturation. The High ISO (ISO6400, ISO3200) and high noise level cases tend to favor the Desaturation module, whereas low ISO (ISO800, ISO1600) and low noise level cases tend to prefer the CCM module, as shown in Figure 5. Because Desaturation can reduce the color noise and saturation, high ISO images with high-level noise prefer it. CCM can remove color casts and enhance color saturation. Therefore, it is the best choice is CCM for low-ISO images.

Subsequently, we observe a divergence between the second and third ISP pipelines during the third stage. While the second ISP pipeline opts for the Tone Mapping module, the third ISP pipeline favors the White Balance module. Upon the analysis of the images subsequent to the second stage, we observe a pronounced color cast problem, particularly prevalent when the overall brightness was relatively high. In such scenarios, the most suitable course of action is to opt for the White Balance module. Conversely, when dealing with images characterized by a high dynamic range, such as those containing electric lights or direct sunlight, it becomes evident that the superior choice is to employ the Tone Mapping module.

Finally, when summarizing the commonalities among the three ISP pipelines, several key conclusions emerge: i) The color correction module notably enhances detection performance (distinct from color correction for image quality tasks). However, optimal choices vary for images with different brightness and noise levels. ii) The Sharpen/Blur module holds a significant position, either enhancing or blurring the image to align with the detection network. iii) Tone Mapping also plays a crucial role,

Table 4: Experimental results considering computational cost on LOD dataset [9]. $\lambda_c = 0.0$ represents the accuracy-oriented, $\lambda_c = 0.1$ stands for efficiency-oriented. The total time represents the average running time of each sample. The efficiency-oriented method has a significant reduction in the average running time for each sample, which is accompanied by a slight decrease in performance. As $\lambda_c$ increases, our method tends to favor faster-executing modules.

| | Exposure | Gamma | CCM | Sharpen Blur | Denoise | Tone Mapping | Contrast | Saturation | Desaturation | White Balance | mAP @0.5 | mAP @0.75 | mAP @0.5:0.95 | Total Time (ms) |
|---|---|---|---|---|---|---|---|---|---|---|---|---|---|---|
| Time (ms) | 1.7 | 2.0 | 1.9 | 6.3 | 10 | 2.7 | 2.1 | 2.0 | 1.9 | 1.7 | - | - | - | - |
| $\lambda_c = 0$ | 0% | 0% | 99.75% | 100% | 0% | 100% | 0.25% | 77.5% | 100% | 22.5% | **71.4** | **51.7** | **47.1** | 14.73 |
| $\lambda_c = 5e-3$ | 4.75% | 1% | 55.5% | 100% | 0% | 59.75% | 43.75% | 39.25% | 100% | 96.5% | 71.1 | 51.5 | 47.0 | 14.30 |
| $\lambda_c = 1e-2$ | 57.5% | 56.25% | 100% | 42.5% | 0% | 42.75% | 37.5% | 21% | 42.5% | 100% | 71.0 | 51.6 | 47.0 | 11.54 |
| $\lambda_c = 5e-2$ | 100% | 7% | 100% | 0% | 0% | 0% | 48% | 49.75% | 95.25% | 100% | 70.0 | 49.9 | 46.0 | 9.26 |
| $\lambda_c = 1e-1$ | 100% | 0.75% | 99.5% | 0% | 0% | 0% | 0% | 99.75% | 100% | 100% | 69.9 | 50.1 | 45.9 | **9.20** |

Table 5: Comparison of object detection performance at different stages on the LOD dataset [9]. Our approach attains optimal performance with just two stages and dynamically achieves a trade-off between object performance and computational cost.

| Methods | Stages | mAP@0.5 | mAP@0.75 | mAP@0.5:0.95 |
|---|---|---|---|---|
| Hyperparameter Optimization [35] | 10 | 70.1 | 49.7 | 46.1 |
| Attention-aware Learning [30] | 10 | 70.9 | 51.0 | 46.6 |
| ReconfigISP [39] | 5 | 69.4 | 49.6 | 45.6 |
| Refactoring ISP [34] | 6 | 68.3 | 47.6 | 44.1 |
| | 1 | 69.6 | 49.8 | 45.7 |
| | 2 | 70.9 | 51.1 | 46.6 |
| Ours | 3 | 71.3 | 50.7 | 46.9 |
| | 4 | 71.4 | 51.6 | 47.1 |
| | 5 | **71.4** | **51.7** | **47.1** |

enhancing detection accuracy by adjusting the overall color and brightness. iv) Denoising, contrary to previous research conclusions, is not deemed crucial. This finding contributes to substantial computational cost savings for the ISP. These analyses and conclusions provide valuable insights for the future designs of ISPs tailored to specific downstream tasks.

**Adaptive Trade-off.** To demonstrate that our model can achieve an adaptive trade-off between efficiency and accuracy during the inference phase, we use stages to represent the number of ISP modules. Recall that none of the previous algorithms support dynamic efficiency-accuracy trade-off. Hyper-parameter Optimization [35] and Attention-aware Learning [30] are optimized based on handcrafted ISP, so the time consumption cannot be changed during the inference phase. Although ReconfigISP [39] and Refactoring ISP [34] have optimized ISP pipelines and parameters for specific tasks, the ISP pipelines and parameters remain fixed during inference. As shown in Table 5, our method only requires 3 stages to achieve the best performance on the LOD dataset and reaches a good trade-off between efficiency and accuracy. Moreover, this trade-off happens at inference time, without any retraining, and supports the dynamic update of the trade-off strategy.

# 5 Conclusion

In this paper, we introduce AdaptiveISP, a novel approach that leverages deep reinforcement learning to automatically generate an optimized ISP pipeline and associated parameters, maximizing detection performance with a pre-trained object detection network. Our method incorporates several key innovations. Firstly, we formulate the ISP configuration process as a Markov Decision Process, allowing reinforcement learning to autonomously discover an optimal pipeline and parameters for specific high-level computer vision tasks. Secondly, to account for computational costs associated with different ISP modules, we introduce a penalty of computational time. Comprehensive experiments demonstrate that AdaptiveISP surpasses existing state-of-the-art methods, and dynamically manages the trade-off between performance and computational cost. Furthermore, we conduct a detailed analysis of individual modules within ISP configurations, offering valuable insights for future ISP designs tailored to specific downstream tasks. The present approach relies on the utilization of differentiable ISP modules in research, with future endeavors aimed at exploring non-differentiable ISP methodologies.

## Acknowledgements

This work is supported by Shanghai Artificial Intelligence Laboratory and RGC Early Career Scheme (ECS) No. 24209224. We also extend our gratitude to Quanyi Li for his insightful discussions and valuable comments.

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

# A  Additional Experiments

## A.1  Experiments Details

We utilize the Adam optimizer with an initial learning rate of $3e-5$ and a batch size of 8. The learning rate gradually decreases by a factor of $\lambda = 0.1^{3 \cdot \text{iter}/\text{iter}_{\text{total}}}$. Note that both the policy network and value network use the same initial learning rate. Our training comprises 100,000 iterations on one NVIDIA RTX 3090 (24G) GPU for the LOD dataset [9], which is completed in around 24 hours. Additionally, we re-implement four methods, namely Hyperparameter Optimization [35], Attention-aware Learning [30], ReconfigISP [39], and Refactoring ISP [34], following the original papers. The ISP pipelines for these methods are illustrated in the RGB domain of Fig. 2 of the main paper. The results for Crafting [9], NeuralAE [27], and DynamicISP [38] are obtained from their respective original papers.

The OnePlus dataset [39] comprises DNG files containing raw images and metadata. Given that our method operates on linear RGB, we employ "rawpy" to perform demosaicing operations on the original raw images, converting them into linear RGB images for the creation of our training and testing datasets.

The LOD [9], OnePlus [39], and raw COCO [19] datasets are commonly used in ISP research. The LOD dataset provides accompanying metadata, which greatly facilitates our experimental analysis. The OnePlus dataset is a real-world dataset collected by smartphones. The COCO dataset is a well-known object detection and segmentation dataset. The ROD [37] dataset is a 24-bit HDR raw dataset collected by the SONY IMX490 sensor. The IMX490 sensor is rare in everyday life, therefore, we do not use ROD as our benchmark dataset.

## A.2  Ablation Experiments on the Necessity of ISP

To validate the necessity of Image Signal Processing (ISP), we conducted a series of experiments on a raw COCO dataset [19]. For these experiments, we randomly selected 5,000 images from the COCO training set as our training dataset. We utilized a pre-trained YOLO-v3 model [32] as our starting model and trained it on the raw COCO dataset, setting the batch size to 128 and the learning rate to 0.01, with the training extending over 100 epochs. Additionally, we trained our method on the same dataset. It is important to note that within our methodology, the YOLO-v3 model [32] remained in a frozen state. The final experimental results were compared on the raw COCO validation set, as detailed in Table 6. These results convincingly demonstrate that the inclusion of the ISP module is crucial and can significantly enhance detection performance.

Table 6: Ablation experiment of the necessity of ISP on raw COCO dataset [19].

| Methods | mAP@0.5 | mAP@0.75 | mAP@0.5:0.95 |
|---|---|---|---|
| Raw + YOLO-v3 [32] | 34.5 | 22.9 | 21.8 |
| Ours | **56.2** | **41.2** | **38.6** |

## A.3  Additional Experiments on ROD dataset

We conduct additional experiments on the ROD dataset. Note that the released ROD dataset differs from the one described in the published paper. Additionally, the released results (AP 28.1) on the new version of the dataset are lower than those reported in the published paper, according to the open-source code released on GitHub, indicating that the released version is more challenging.

Because the released dataset is only a training dataset that provides paired raw images and annotations, we randomly split 80% of the dataset (12,800) for training, with the remainder as our validation dataset (3200). The dataset processing pipeline is similar to the original paper and released codes. Since our method emphasizes using training-well models, we selected only three categories (person, car, truck) belonging to COCO from the ROD dataset for a fair comparison. Due to time constraints, we select the previously best-performing method, Attention-Aware Learning [30], and the state-of-the-art method on the ROD dataset, Toward RAW Object Detection [37], as our comparison methods. Each method was trained for 100 epochs.

As shown in Table 7, our method achieves the best performance, even though the detector we used is not trained on this input (Toward RAW Object Detection method [37] does).

Table 7: Experimental results on ROD dataset. * refers to a detector that is trained on raw input, which is normally better than detectors only trained on RGB input (like ours). The "Toward RAW Object Detection" is an end-to-end raw detection method, that updates its parameters during training time. Other methods use a pre-trained YOLO-v3 detector and freeze its parameters during training time.

| Methods | mAP@0.5 | mAP@0.75 | mAP@0.5:0.95 |
|---|---|---|---|
| Attention-Aware Learning [30] | 49.8 | 34.0 | 31.6 |
| Toward RAW Object Detection* [37] | **54.1** | 30.9 | 31.3 |
| Ours | 51.6 | **35.7** | **33.2** |

## A.4 Ablation Experiments on Penalty of Reusing

Table 8: Ablation experiment of the penalty of reusing on LOD dataset [9].

| Penalty of Reusing | Exposure | Gamma | CCM | Sharpen Blur | Denoise | Tone Mapping | Contrast | Saturation | Desaturation | White Balance | mAP @0.5 |
|---|---|---|---|---|---|---|---|---|---|---|---|
|  | 2.25% | 0% | **164%** | **139%** | 0% | 46.5% | 0% | 0% | 99.5% | 48.75% | 70.9 |
| ✓ | 0% | 0% | 99.75% | 100% | 0% | 100% | 0.25% | 77.5% | 100% | 22.5% | **71.4** |

To assess the impact of the penalty of reusing, we perform ablation experiments on the LOD dataset [9]. As shown in Table 8, if the penalty of reusing is not applied, there is a noticeable decline in detection performance, accompanied by an occurrence frequency exceeding 100% for both CCM and Sharpen/Blur modules.

## A.5 Experiment on Limited Datasets

To validate the applicability of our method with limited training data, we conducted experiments on the OnePlus dataset [39], which comprises only 50 training images. The results are presented in Table 9, demonstrating that our approach outperforms ReconfigISP.

Table 9: Experimental results on limited datasets (OnePlus dataset [39]).

| | mAP@0.5 | mAP@0.75 | mAP0.5:0.95 |
|---|---|---|---|
| ReconfigISP [39] | 60.1 | - | - |
| Ours | **75.6** | **50.5** | **47.3** |

## A.6 Comparison with Image Quality Task

To validate the distinct requirements of ISP between image quality tasks and object detection tasks, we select Exposure [10] as the representative method for image quality tasks. Given that the Exposure [10] method requires paired raw-RGB data, and computing object detection results necessitate bounding box labels, we utilize 1,000 simulated raw-like images with the most bounding boxes converted from the COCO training dataset [19] using UPI [1] as our training dataset. Additionally, we employ the LOD Dataset, comprising all real raw images, as our test dataset, similar to prior works [9, 38, 30, 31]. We train both our approach and the Exposure method on the synthetic raw COCO dataset.

As shown in Figure 6, image quality and object detection tasks have distinct requirements for ISP. Image quality tasks primarily emphasize color and brightness, intending to produce images that closely align with human perception. In contrast, the results obtained from processing images for target detection tasks better meet the demands of machine-based systems. We further compare the object detection results with the image quality method [10] on the LOD dataset with all real raw images, as shown in Table 10, the results of our method are much higher than the image quality method.

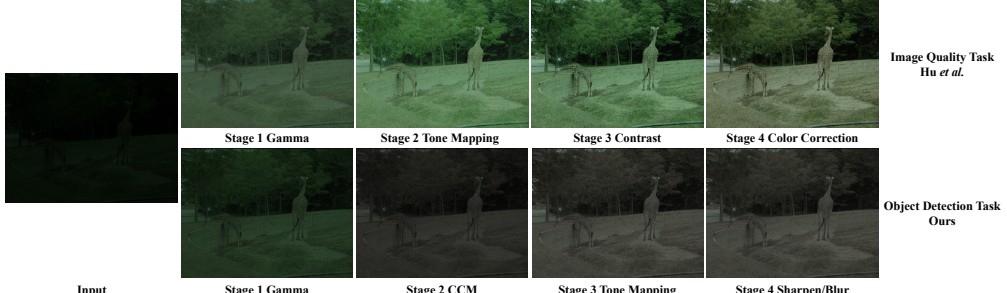

Figure 6: Visualization results for image quality and object detection tasks on the raw COCO dataset. Image quality tasks and object detection tasks have distinct requirements for ISP.

Table 10: Comparison results with image quality methods on all LOD datasets.

|  | mAP@0.5 | mAP@0.75 | mAP@0.5:0.95 |
|---|---|---|---|
| Hu *et al.* [10] | 54.3 | 35.4 | 33.7 |
| Ours | **62.1** | **42.1** | **39.6** |

## A.7 More Visualization Results

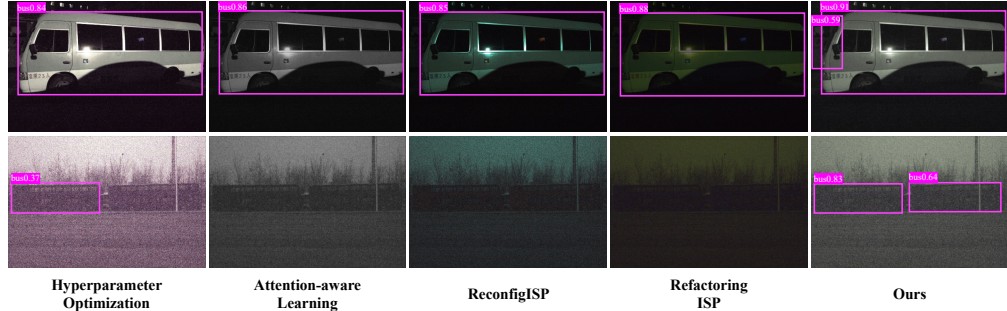

Figure 7: Object detection visualization results on LOD datasets.

We show more object detection and image segmentation visualization results in Figure 7 and Figure 8, showcasing its superior performance in terms of both missed detection and false detection compared to all other methods.

## B  Details of Implemented ISP

As shown in Figure 9, a typical camera ISP pipeline includes two parts: raw-domain and RGB-domain. The raw-domain processing converts a raw sensor signal to a linear RGB image, which includes Defective Pixel Correction (DPC), Black Level Correction (BLC), Lens Shading Correction (LSC), and Demosaicking. The RGB-domain processing further applies customized rendering and post-processing to generate the final image. This processing includes tone mapping, color correction, denoising, sharpening, etc. We focus on RGB-domain processing, the ISP detailed in this paper encompasses various standard differentiable modules, including:

(1) Exposure: Exposure control regulates the amount of light reaching the sensor to ensure the image's overall brightness matches the desired level. This module helps in adapting to varying lighting conditions. This module can be expressed as:

$$I_{exposure} = I \cdot 2^p, \tag{12}$$

where $I$ is the input image, $p$ is the exposure parameter and $p \in [-3.5, 3.5]$.

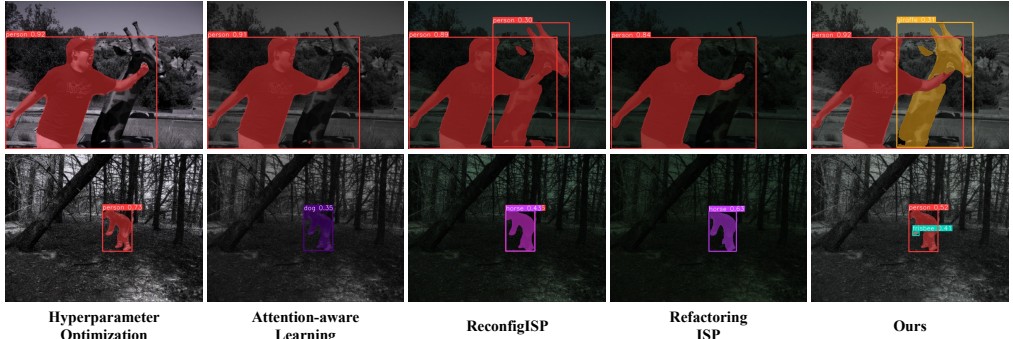

| Hyperparameter Optimization | Attention-aware Learning | ReconfigISP | Refactoring ISP | Ours |

Figure 8: Image segmentation visualization results on raw COCO datasets.

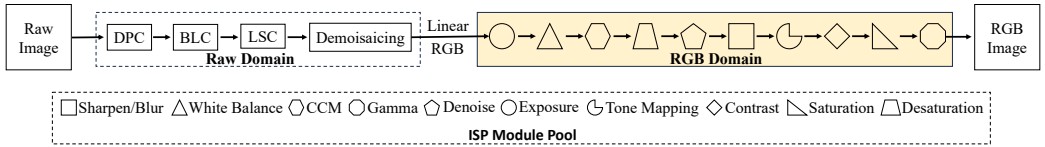

Figure 9: A typical camera ISP pipeline consists of different modules in the raw domain and RGB domain, which transforms raw sensor pixel data into RGB images suitable for viewing. In this paper, we focus on the ISP modules in the RGB domain.

(2) White Balance: White balance adjustment aims to rectify color temperature discrepancies, ensuring that colors in the final image appear natural and consistent with the observed scene. This module can be expressed as:

$$\begin{bmatrix} I_{R'} \\ I_{G'} \\ I_{B'} \end{bmatrix} = \begin{bmatrix} p_R & 0 & 0 \\ 0 & p_G & 0 \\ 0 & 0 & p_B \end{bmatrix} \begin{bmatrix} I_R \\ I_G \\ I_B \end{bmatrix}, \tag{13}$$

where $p_R, p_G, p_B$ are gain value of input image $I_R, I_G, I_B$ and $p_R, p_G, p_B \in [e^{-1/2}, e^{1/2}]$.

(3) Color Correction Matrix (CCM): The CCM module fine-tunes color representations by mapping the sensor's color response to the desired color space. It plays a pivotal role in achieving accurate color rendering, the original image data $I_R, I_G, I_B$ is multiplied with CCM to obtain $I_{R'}, I_{G'}, I_{B'}$:

$$\begin{bmatrix} I_{R'} \\ I_{G'} \\ I_{B'} \end{bmatrix} = \begin{bmatrix} p_{00} & p_{01} & p_{02} \\ p_{10} & p_{11} & p_{12} \\ p_{20} & p_{21} & p_{22} \end{bmatrix} \begin{bmatrix} I_R \\ I_G \\ I_B \end{bmatrix}, \tag{14}$$

where $p$ is the CCM parameter and the sum of elements in each row equals 1.

(4) Gamma: Gamma correction alters the image's tonal curve, affecting the distribution of lightness levels. This module is essential for ensuring luminance and contrast consistency. This module can be expressed as:

$$I_{gamma} = I^p, \tag{15}$$

where $p$ is the gamma parameter and $p \in [1/3, 3]$.

(5) Denoise: Denoise techniques are applied to mitigate image noise, particularly in low-light or high ISO settings, preserving image quality by reducing noise. In this paper, we implement a differentiable Non-Local Means Denoising algorithm (NLM [3]) based on PyTorch. This module can be expressed as:

$$I_{denoise} = \text{NLM}(I, p), \tag{16}$$

where $p \in [0, 1]$ is filter-strength, which represents the strength of denoising.

(6) Sharpen/Blur: This module enhances or blurs image details and edges, thus improving image clarity and visual appeal. It is useful for accentuating finer image structures. This module can be expressed as:

$$I_{sharpen/blur} = p \cdot I + (1 - p) \cdot I_{blurred}, \tag{17}$$

where $p \in [0, 2]$ is the factor and $I_{blurred}$ is the blurred image, which can be obtained by

using the blur kernel, and the blur kernel can be expressed as: $1/13 \cdot \begin{bmatrix} 1 & 1 & 1 \\ 1 & 5 & 1 \\ 1 & 1 & 1 \end{bmatrix}$.

(7) Tone Mapping: Tone mapping regulates the dynamic range of an image, ensuring that details in both bright and dark areas are discernible. It is especially valuable for rendering high dynamic range scenes on standard displays. Following [10], we approximate curves as monotonic and piecewise-linear functions. We use $L$ parameters represent tone mapping curve, denoted as $\{p_0, p_1, \ldots, p_{L-1}\}$. With the prefix-sum of parameters defined as $P_k = \sum_{l=0}^{k-1} p_l$, the points on the curves are represented as $(k/L, P_k/P_L)$. Given an input image $I \in [0, 1]$, the mapped result can be expressed as:

$$I_{tone\_mapping} = \frac{1}{P_L} \sum_{i=0}^{L-1} \text{clip}(L \cdot I - i, 0, 1) p_k, \tag{18}$$

where the slope of each segment in the curve is in $[0.5, 2.0]$ and $L = 8$.

(8) Contrast: This module modulates the contrast of an image, defining the variations between light and dark areas. It contributes to image aesthetics and enhances perceptual quality. This module can be expressed as:

$$I_{contrast} = (1 - p) \cdot I + p \cdot I \cdot \frac{\frac{1}{2}(1 - \cos(\pi \cdot I_{lum}))}{I_{lum}}, \tag{19}$$

where $p \in [-1, 1]$ is adjustment factor, and $I_{lum}$ can be expressed as:

$$I_{lum} = 0.27 \cdot I_R + 0.67 \cdot I_G + 0.06 \cdot I_B. \tag{20}$$

(9) Saturation: Saturation adjustment manipulates the intensity of colors in an image, allowing for vivid or muted color representation according to the desired artistic effect. This module can be expressed as:

$$\begin{aligned} (I_H, I_S, I_V) &= \text{RgbToHsv}(I_R, I_G, I_B), \\ I_{S'} &= I_S + (1 - I_S) \cdot (0.5 - |0.5 - I_V|) \cdot 0.8, \\ I' &= \text{HsvToRgb}(I_H, I_{S'}, I_V), \\ I_{saturation} &= (1 - p) \cdot I + p \cdot I \cdot \frac{\frac{1}{2}(1 - \cos(\pi \cdot I'))}{I_{lum}}, \end{aligned} \tag{21}$$

where $p \in [0, 1]$ is adjustment factor, and $I_{lum}$ can be obtained through Equation 20.

(10) Desaturation: The desaturation module, conversely, reduces the intensity of colors in an image, leading to a more muted or grayscale appearance. It is often employed for specific artistic or visual effects, including creating black-and-white imagery or subtle color emphasis. This module can be expressed as:

$$I_{desaturation} = (1 - p) \cdot I + p \cdot (I_{lum}, I_{lum}, I_{lum}), \tag{22}$$

where $p \in [0, 1]$ is adjustment factor, $(I_{lum}, I_{lum}, I_{lum})$ represents R-G-B channels of the image are $I_{lum}$, and $I_{lum}$ can be obtained through Equation 20.

## C  Additional Details of Method

### C.1  Network Architecture

All of these networks share the same architecture depicted in Figure 10, while the extra channel (EC) represents different additional input information, more detail can be found in Fig. 4 of the main paper. For each feature extraction, we use four Conv-BN-LRelu layers and a fully connected layer with 128 dimensions. Each Conv-BN-LRelu layer consists of a convolution layer with $4 \times 4$ kernels and $2 \times 2$ strides, a batch normalization, and an LRelu activation of $leak = 0.2$, the first layer has 32 channels, and this number progressively doubles in the subsequent layers. The policy network includes a dropout layer (with a dropout rate of $p = 0.5$) preceding the fully connected layer

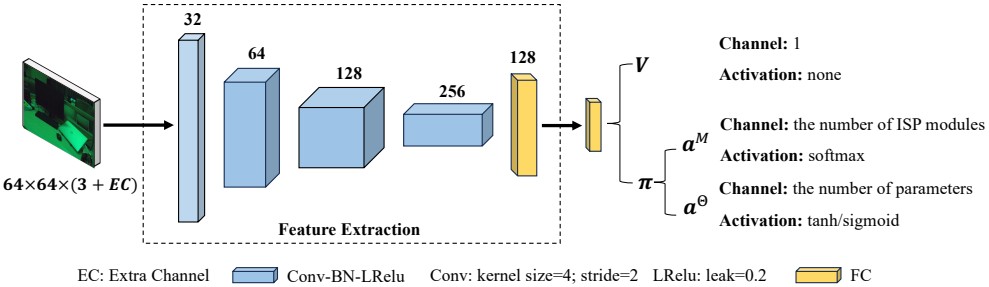

Figure 10: The network architecture of the policy and value network in our method. The extra channel (EC) represents additional input that needs to be supplemented.

with 128 dimensions. The activation function for the module selection network is softmax, with the number of outputs from the module selection network corresponding to the number of ISP modules. Parameter prediction networks share a common feature extraction backbone. The activation function and the number of outputs from parameter prediction networks are specific to each module, reflecting variations in both the number and range of parameters for each ISP module.

## C.2 Terminated and Truncated

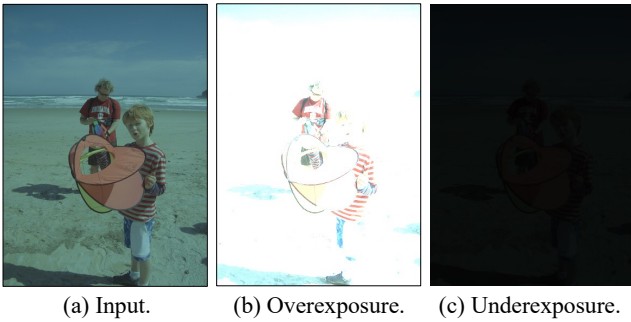

(a) Input.     (b) Overexposure.     (c) Underexposure.

Figure 11: Failure cases during the training stage. When the system predicts an excessively high gain value for the exposure module, the resulting image becomes overexposed, as illustrated in (b). Conversely, an overly small gain value leads to a completely black image, as depicted in (c).

The process of reinforcement learning can be abstracted as an agent continually gathering experiences from the environment and learning from them. However, during the experience collection process, issues such as termination and truncation can significantly impact experience collection and training, necessitating special design considerations.

We can limit the total number of ISP configuration steps to enable the policy network to autonomously learn the optimal module selection for downstream tasks under resource constraints. Specifically, the number of ISP module options in our ISP module pool is denoted as N, for the policy network selecting the modules more stably, we intentionally "terminated" once the execution stage reaches the maximum $T$ ($T \leq N$). Besides, we incorporate step information into the policy networks by introducing an additional channel for providing stage-related data, as shown in the stage channel of Fig. 4 of the main paper. The terminated coefficient $\lambda_{terminated}$ can be defined as:

$$\lambda_{terminated} = \begin{cases} 0, & \text{if } t \leq T, \\ 1, & \text{otherwise}, \end{cases} \tag{23}$$

where $t$ is the number of ISP configuration stages, and $T$ is the maximum stages and $T$ is 5 in our experiments.

Additionally, we intentionally truncate the execution when the network diverges to output abnormal images, as shown in Figure 11, for example, "truncated" once the image mean gets out of range

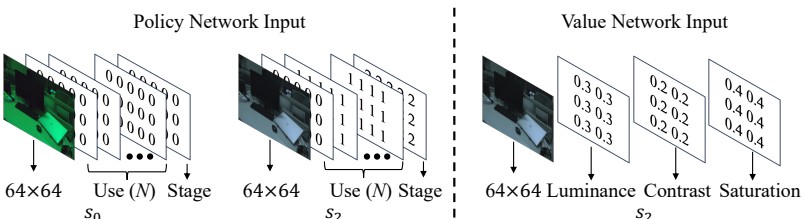

Figure 12: The network input of the policy and value network in our method. The number of use channels is the same as the number of ISP modules ($N$). The stage channel represents the current stage. The extra 3 channels of value network input respectively represent the luminance, contrast, and saturation of the input image.

($lum_{min}$, $lum_{max}$). The truncated coefficient $\lambda_{truncated}$ can be defined as:

$$\lambda_{truncated} = \begin{cases} 0, & \text{if } lum_{min} \leq \overline{I} \leq lum_{max}, \\ 1, & \text{otherwise }, \end{cases} \tag{24}$$

where $\overline{I}$ is the mean of processed image.

Ultimately, the state value function $V^\pi(s)$ can be defined as $V^\pi(s) = (1 - \lambda_{terminated}) \times (1 - \lambda_{truncated})V^\pi(s)$.

### C.3   Design for Object Detection

**ISP Modules.**  In the ISP pipelines for visual quality [10, 16, 28], the ISP modules primarily focused on pixel operations related to color and brightness adjustment.  To better adapt the ISP pipeline for object detection, following the traditional ISP modules, we extend the existing pipeline by incorporating differentiable Sharpen/Blur, Denoise, and CCM modules onto the foundation established in [10]. Empirically, we find these changes increase the detection performance.

**Input Resolution.** For object detection networks, larger pixel images are commonly employed for training and testing [4, 33, 8, 21, 18, 5]. To seamlessly integrate our policy network and detection network into end-to-end training, the input resolution to our ISP is configured based on the input resolution of the detection network. This ensures the back-propagation from the detection network to the differentiable ISP and subsequently to the policy network. The policy network and value network can continue to utilize down-sampled input resolutions, such as $64 \times 64$ pixels, during training and testing, as illustrated in Figure 12. To enhance the value network's assessment of the current state, we include the average luminance, contrast, and saturation of the input image as additional input features.

