# OpenReview forum: "AdaptiveISP: Learning an Adaptive Image Signal Processor for Object Detection"
_NeurIPS.cc/2024/Conference — NeurIPS 2024 poster_

### Official Review · Reviewer_bosW · 2024-07-09

**Soundness:** 3
**Presentation:** 3
**Contribution:** 3
**Rating:** 7
**Confidence:** 4

**Summary:**

Image Signal Processors (ISP) are software pipelines that aim to improve images for their visual quality or application-dependent downstream tasks. This work presents AdaptiveISP, a method to simultaneously optimize an ISP pipeline, consisting of individual functions such as image sharpening or color correction and the functions' parameters. Learning policies that map an image into the optimal ISP structure and parameters while considering computation cost and the specific downstream task improves performance compared to the prior art and allows for real-time application, adapting to newly shot images in time.

**Strengths:**

- When creating novel ML methods, it is important to consider not only accuracy on pre-recorded datasets but also how these methods can be applied in the field, including adaptations of parameters and cost of computation. This work tackles these questions concerning image signal processing tasks, a significant endeavor for any robotic system equipped with a camera.
- The presented method incorporates the specific downstream task, e.g., object detection, together with an adaptive trade-off for computation time, providing a comprehensive framework that has been presented with high clarity and incorporates a good level of originality, e.g., not aiming for visual clarity but seeking to obtain images best suited for the employed network.

**Weaknesses:**

- Since real-time applicability is presented as a key motivation for the paper, this work could benefit from an expanded evaluation of the method's computational flexibility. For example, Table 3 shows the average running time for two settings of the $\lambda_c$ parameter. It would be more informative to see the results for a range of values for $\lambda_c$ instead, not just showing that the expected impact is achieved but what the limitations and behavior of the system are when trying to tune for accuracy or speed. Similarly, one could experimentally consider how the method's time and memory demands change with varying pools of ISP modules to choose from.

**Questions:**

- Figure 1 shows how different ISP modules are placed in the pipeline to modify the image's color values. Have you considered further calibration steps to be part of the adaptive pipeline, e.g., corrections for fish eye effects and other optical deficiencies?

**Limitations:**

The authors have adequately addressed the limitations.

---

> ### Author Rebuttal · Authors · 2024-08-07
>
> Thank you for your feedback, we provide the results for a range of values for 𝜆𝑐, as shown in Table 3 on the global PDF, the efficiency-oriented method significantly reduces the average running time for each sample, with only a slight decrease in performance. As 𝜆𝑐 increases, our method tends to favor faster-executing modules. We will consider memory demands in the future. And we will consider including the calibration module as part of the adaptive pipeline in the future.

---

### Official Review · Reviewer_z6yj · 2024-07-11

**Soundness:** 3
**Presentation:** 3
**Contribution:** 3
**Rating:** 5
**Confidence:** 4

**Summary:**

This paper proposes AdaptiveISP, a task-driven and scene-adaptive ISP, which uses deep reinforcement learning to automatically generate an optimal ISP pipeline and associated ISP parameters, aiming to maximize the detection performance. Experimental results show that AdaptiveISP outperforms prior state-of-the-art methods for object detection, and it effectively manages the trade-off between detection performance and computational cost, demonstrating its effectiveness in dynamic scenes.

**Strengths:**

1. The AdaptiveISP pipeline combines ISP and detection together and turns the fixed process into a task-oriented tuning problem, which demonstrates greater potential for specific tasks.
2. The results of the algorithm on some datasets look good.

**Weaknesses:**

1. The experiments are performed only on YOLO detectors. The conclusion and findings could be more solid and convincing if experiments are available on other architectures.
2. It would be better to analyze and validate the generalization ability with some 3rd-party datasets.

**Questions:**

1. The tuning process could be costly when applied to real scenes since there are more complex factors in real scenes than in the experimental datasets. For example, the light condition can change rapidly in urban scenes during the nighttime. It is doubtful whether adaptiveISP can react to the change.
2. Is it stable to tune adaptiveISP using RL? Will the tuning lead to even worse results than traditional ISPs? How can you evaluate the risk of your work in applications.?

**Limitations:**

See weakness

---

> ### Author Rebuttal · Authors · 2024-08-07
>
> 1. **Performance on a newer detector.**  Please refer to part 1 of the Author Rebuttal.
>
> 2. **Generalization ability.** We utilized a model trained on the LOD dataset and conducted testing on the OnePlus dataset. As shown in Table 1 of the main paper, our method demonstrates the best generalization ability.
>
> 3. **The tuning process is cheap.** Our method tunes an ISP pipeline in just 20ms (6ms for the tuning process + 14ms for ISP execution). We believe it can quickly adapt to rapidly changing light conditions, and the tuning process is efficient enough for ISP processing.
>
> 4. **The stability of adaptiveISP.** We selected the worst 10% of samples from our method on the LOD validation dataset and tested them using the traditional ISP method. Our method achieved 35.6 mAP@50:95, while traditional ISPs only achieved 24.0 mAP@50:95, demonstrating that our method is stable.

---

> > ### Comment · Reviewer_z6yj · 2024-08-13
> > **After Rebuttal**
> >
> > The rebuttal has addressed most of my concerns. I will keep my initial rating. I strongly recommend the authors to add more discussions on image quality tasks, as pointed out by Reviewer Ln32, which is particularly important for an ISP.

---

> > > ### Author Response · Authors · 2024-08-13
> > >
> > > Thank you for your response. As discussed in our paper, the requirements for ISP differ significantly between high-level computer vision tasks and image quality tasks. Our primary focus is on optimizing ISP specifically for advanced computer vision tasks, which is critically important in scenarios such as autonomous driving.

---

### Official Review · Reviewer_Sot8 · 2024-07-12

**Soundness:** 3
**Presentation:** 3
**Contribution:** 2
**Rating:** 5
**Confidence:** 4

**Summary:**

This paper proposes a novel approach to image signal processing (ISP) specifically tailored for object detection tasks, leveraging deep reinforcement learning to optimize both ISP structures and parameters. This method dynamically adjusts the ISP pipeline in response to different scene requirements, which enhances detection performance.

**Strengths:**

1.	The figures in this paper are of good quality and easy to understand.
2.	AdaptiveISP can dynamically adjust the ISP pipeline according to different input images to adapt to different scene changes.

**Weaknesses:**

1.	The system's performance heavily relies on the quality of the pre-trained object detection models. There is a potential limitation in cases where these models do not generalize well or when transitioning to different object detection tasks that were not part of the initial training set.
2.	The challenges and contributions are too general and not prepared objectively. It should briefly highlight the paper's novelty as what is the main problem, how has it been resolved and where the novelty lies?
3.	Although the paper conducted experiments on multiple datasets, the limited diversity and coverage of these datasets may not be sufficient to fully validate the performance of AdaptiveISP in various real-world application scenarios. For example, there is a lack of testing on the DAWN dataset, a dataset that covers multiple weather scenarios and is well-suited for dynamic testing.

**Questions:**

The author mentions "Our method only takes 1.2 ms per stage during inference" , what does "per stage" mean, is it the end-to-end inference time for each image?

**Limitations:**

YOLOv3 is an older model, how does the method in this paper perform under newer YOLO models or other models? It is suggested that the authors add other models to the experiment.

---

> ### Author Rebuttal · Authors · 2024-08-07
>
> 1. **The performance of different detectors.** Please refer to part 2 of the Author Rebuttal.
>
> 2. **Briefly highlight the paper's novelty.**
>
>      We aim to design the first adaptive ISP tailored for detection. There are two main challenges: complexity and efficiency. First, optimizing ISP modules, updating their parameters, and enhancing downstream recognition is a complex task, so prior works have only updated parameters. Second, ISP optimization must be efficient enough for real-time applications like autonomous driving and robotics, but most methods rely on searching strategies, making them impractical for dynamically changing scenes.
>
>     To solve these challenges, we proposed three main innovations: (1) we model ISP configuration as a Markov Decision Process, integrate a pre-trained detector, and design a real-time reconfigurable ISP system based on reinforcement learning; (2) we introduce a new cost penalty mechanism, enabling AdaptiveISP  to dynamically trading off object detection accuracy and ISP latency; (3) we analyze the ISP pipeline predicted by our method to provide some insights for future ISP design work.
>
> 3. **The performance of AdaptiveISP in various real-world application scenarios.** We conduct additional experiments on real-world HDR raw datasets, our method achieves the best performance, please refer to part 2 of the Author Rebuttal. Since DAWN is not a raw image dataset, which may not fit into our setup, we only test the ROD dataset.
>
> 4. **The meaning of "per stage".**  The ISP comprises multiple modules, such as exposure, white balance, and gamma correction. Each running ISP module represents a stage. We model the ISP configuration process as a Markov Decision Process, allowing our method to sequentially predict the ISP’s modules at inference time.
>
> 5. **Performance on a newer detector.**  Please refer to part 1 of the Author Rebuttal.

---

> > ### Comment · Reviewer_Sot8 · 2024-08-13
> >
> > The rebuttal has addressed most of my concerns. I will retain my initial rating.

---

### Official Review · Reviewer_Ln32 · 2024-07-13

**Soundness:** 2
**Presentation:** 2
**Contribution:** 2
**Rating:** 3
**Confidence:** 5

**Summary:**

A new perspective of designing ISP pipeline. Good results with some problems should be addressed.

**Strengths:**

1/ One method for raw detection which is still a new subarea waiting more discovery.
2/ Good performance compared with some methods.
3/ Discuss some orders in ISP pipeline.

**Weaknesses:**

1/ Discussion on related works especially for ISP pipeline is not sufficient. Not only the ISP for task performance [1], but also for image quality [2,3].
2/ The datasets used for comparison are not actually real raw detection data. It is doubtful whether the real scene performance is good.
3/ Also, the compared methods are for designing special ISP params or orders to get better performance on  raw downstream tasks. However, what about using an existing ISP and detect on RGB images? For datasets such as LOD and COCO can do it. The results must be better than existing RGB detection SOTAs, so that this work can be meaningful.
4/ The order of ISP modules is still not fixed. There have been many works finding that it can be various and change its order according to tasks or image quality. Also, the pipelines can be different according to manufacturers.
5/ Some new benchmarks such as [1] should be used for evaluation.

[1] Ruikang Xu, Chang Chen, Jingyang Peng, Cheng Li, Yibin Huang, Fenglong Song, Youliang Yan, Zhiwei Xiong: Toward RAW Object Detection: A New Benchmark and A New Model. CVPR 2023: 13384-13393
[2] Woohyeok Kim, Geonu Kim, Junyong Lee, Seungyong Lee, Seung-Hwan Baek, Sunghyun Cho: ParamISP: Learned Forward and Inverse ISPs using Camera Parameters. CoRR abs/2312.13313 (2023)
[3] Syed Waqas Zamir, Aditya Arora, Salman H. Khan, Munawar Hayat, Fahad Shahbaz Khan, Ming-Hsuan Yang, Ling Shao: CycleISP: Real Image Restoration via Improved Data Synthesis. CVPR 2020: 2693-2702

**Questions:**

Please refer to the weaknesses section.

**Limitations:**

1/ Lack of new benchmark, new methods.
2/ Must compare with RGB SOTAs.

---

> ### Author Rebuttal · Authors · 2024-08-07
>
> 1. **Related works, especially for ISP pipelines, are not sufficient.** In this paper, we primarily discuss how to design an ISP tailored for a specific high-level computer vision task. In most ISP tuning or design topics, the ISP consists of multiple modules with distinct roles. Using a network to mimic an end-to-end whole ISP or one of the modules of an ISP for computer vision or image quality tasks significantly differs from the objectives discussed in our paper. In addition, we would be happy to include these related works in the final version of our paper.
>
> 2. **The datasets used for comparison are not real raw detection data.** The LOD and OnePlus datasets are real raw detection datasets collected in the real world. Since the demosaicking module lacks parameters and does not alter the number of channels, most ISP-related research tasks, including the mentioned paper "Toward RAW Object Detection", use post-demosaicking results as input.
>
> 3. **Comparison with RGB (existing ISP) detection SOTAs.**  Please refer to part 1 of the Author Rebuttal.
>
> 4. **New benchmarks.** Please refer to part 2 of the Author Rebuttal.
>
> 5. **Many works are finding that ISP modules can change their order according to  computer version tasks or image quality.**  In the introduction and related work sections, we mention works such as [9, 23, 29, 33], which demonstrate that ISP modules can adapt their order for specific computer vision tasks or to enhance image quality. Specifically, methods [9, 23] optimize ISPs for image quality, method [29] focuses on object detection tasks, and method [33] addresses both image quality and object detection tasks.
>
> 6. **The pipelines can be different.** This is one of the novelty of our method. As described in Section 4.2 of the main paper,  there are different pipeline requirements in different scenarios.

---

### Author Rebuttal · Authors · 2024-08-07

We thank the reviewers for their feedback. We will revise the manuscript as suggested. Below are responses to common questions. We hope this can address your concerns. If you have other concerns, we will reply as soon as possible.

1. **Additional experiments on different detectors and comparison with RGB (existing ISP) detection SOTAs.** (Reviewer Ln32, Sot8, z6yj)

    We use the detection results from the RGB (existing ISP) as a baseline and conduct comparative experiments on DDQ [1] and YOLOX [2] detectors. As shown in Table 1 of the global PDF, all detectors using our AdaptiveISP demonstrate improved detection performance, demonstrating that our method does not overfit one detector, but is suitable for other detectors. It is important to note that DDQ and YOLOX are not used in the training process, but our ISP can still generalize to these detectors at testing time.

    [1] Zhang, Shilong, et al. "Dense distinct query for end-to-end object detection." CVPR 2023.

    [2] Ge, Zheng, et al. "YOLOX: Exceeding yolo series in 2021." arXiv 2021.

2. **New benchmark or evaluation dataset.** (Reviewer Ln32, Sot8, z6yj)

      We conduct new experiments on the ROD dataset. As shown in Table 2 of the global PDF, our method achieves the best performance, even though the detector we used is not trained on this input (Toward RAW Object Detection method does).

      The LOD, OnePlus, and raw COCO datasets are commonly used in ISP research. The LOD dataset provides accompanying metadata, which greatly facilitates our experimental analysis. The OnePlus dataset is a real-world dataset collected by smartphones. The COCO dataset is a well-known object detection and segmentation dataset. The ROD dataset is a 24-bit HDR raw dataset collected by the SONY IMX490 sensor.  The IMX490 sensor is rare in everyday life; therefore, we do not use ROD as our benchmark dataset.

      Following the reviewers' suggestions, we also conducted comparison experiments on the ROD dataset. Note that the released ROD dataset differs from the one described in the published paper. Additionally, the released results (AP 28.1) on the new version of the dataset are lower than those reported in the published paper, according to the open-source code released on GitHub, indicating that the released version is more challenging.

    Because the released dataset is only a training dataset that provides paired raw images and annotations, we randomly split 80% of the dataset (12,800) for training, with the remainder as our validation dataset (3200). The dataset processing pipeline is similar to the original paper and released codes. Since our method emphasizes using training-well models, we selected only three categories (person, car, truck) belonging to COCO from the ROD dataset for a fair comparison. Due to time constraints, we select the previously best-performing method, Attention-Aware Learning, and the state-of-the-art method on the ROD dataset, Toward RAW Object Detection, as our comparison methods. Each method was trained for 100 epochs.

---

### Decision · Program_Chairs · 2024-09-25

**Decision:**

Accept (poster)

**Comment:**

The paper presents an approach to image signal processing (ISP) tailored for object detection tasks, leveraging deep reinforcement learning to optimize both ISP structures and parameters. The work introduces AdaptiveISP, which dynamically adjusts the ISP pipeline in response to varying scene requirements, enhancing detection performance while managing computational costs. The reviews are mixed, with some highlighting the paper's potential and others expressing concerns about its limitations. The submission received mixed ratings, namely, one accept, two borderline accept, and one reject. For the reviewer voting for rejection, the reviewer did not provide feedback on author rebuttal. The AC checked all the materials, and believes the rebuttal is reasonable. Thus, the AC decides to accept the paper based on the review comments of the other three reviewers.

However, the concerns raised regarding the evaluation scope, related work discussion, and generalization need to be addressed in more detail. The paper would benefit from a revision that includes a broader evaluation across different models and datasets, as well as a more comprehensive discussion of related work.